# High-Performance Photocatalytic Cementitious Materials Containing Synthetic Fibers and Shrinkage-Reducing Admixture

**DOI:** 10.3390/ma13081828

**Published:** 2020-04-13

**Authors:** Jung-Jun Park, Soonho Kim, Wonsik Shin, Hong-Joon Choi, Gi-Joon Park, Doo-Yeol Yoo

**Affiliations:** 1Department of Infrastructure Safety Research, Korea Institute of Civil Engineering and Building Technology, 283 Daehwa-dong, Goyangdae-ro, Ilsanseo-gu, Goyang-si, Gyeonggi-do 10223, Korea; jjpark@kict.re.kr (J.-J.P.); joon7767@kict.re.kr (G.-J.P.); 2Department of Architectural Engineering, Hanyang University, 222 Wangsimni-ro, Seongdong-gu, Seoul 04763, Korea; tnsgh0905@hanyang.ac.kr (S.K.); swonsik214@hanyang.ac.kr (W.S.); spstarg@hanyang.ac.kr (H.-J.C.)

**Keywords:** cement mortar, TiO_2_ power, white Portland cement, NO*_x_* removal, synthetic fibers, material properties

## Abstract

This study aims to examine the mechanical, shrinkage and chemical properties of photocatalytic cementitious materials containing synthetic fibers and a shrinkage-reducing admixture (SRA). Two types of titanium dioxide (TiO_2_) powders and white Portland cement were considered along with ordinary Portland cement (OPC) as a control. Two types of synthetic fibers, i.e., glass and polyethylene (PE), and an SRA with contents varying from 0% to 3% were also considered. Using the TiO_2_ powders and the white Portland cement was effective in reducing the nitrogen oxides (NO*_x_*_)_ concentration in cement composites. The use of PE fibers was more effective than glass fibers in terms of the mechanical properties, i.e., the compressive strength and tensile performance. With the addition of TiO_2_ powders and SRA or the replacement of OPC with white cement, the mechanical properties of the cement mortar generally deteriorated. The total shrinkage of the mortar could be reduced by incorporating the fibers at volume fractions greater than 1%, and the glass fiber was more effective than the PE fiber in this regard. The TiO_2_ powders had no significant impact on the shrinkage reduction of the cement mortar, whereas the SRA and the white Portland cement effectively reduced shrinkage. The addition of 3% SRA decreased the total shrinkage by 43%, while the replacement of the OPC with white cement resulted in a 20% reduction in the shrinkage.

## 1. Introduction

Various harmful gas emissions, which are increasing particularly in large cities around the world, pose a high risk to human health. It is thus necessary to remove or decrease the emissions of substances such as nitrogen oxides (NO*_x_*), sulfur oxides (SO*_x_*) and volatile organic compounds (VOCs). Civil and architectural engineers have attempted to remove these pollutants using extensively exposed building surfaces based on the concept of photocatalytic reactions: the Jubilee church, Italy; the Air France headquarters, France; and pavements in Japan and in the US [1]. The mechanism of NO*_x_* removal by photocatalytic cementitious material has been established: titanium dioxide (TiO_2_) absorbs photons mainly from ultraviolet (UV) light and generates electron ($e_{\mathrm{CB}}^{-}$) and hole ($h_{\mathrm{VB}}^{+}$) pairs. From water and oxygen in an atmosphere, a hydroxyl radical (·OH) and superoxide ($O_{2}^{-}$) can be formed to transform the NO*_x_* into a nitrate ion (${\mathrm{NO}}_{3}^{-}$) that can be removed from the surface of the material, typically by rain [2,3,4].

Based on an experimental study of the self-cleaning of organic dyes and NO*_x_* degradation, Jimenez-Relinque et al. [5] found that (1) an addition of 2% TiO_2_ by weight of cement decreases the workability and compressive strength of the mortar; (2) the photocatalytic efficiency for both NO_x_ and self-cleaning occurs in descending order of ordinary Portland cement, fly ash and blast furnace slag mortars; and (3) the efficiency of the degradation of NO*_x_* increases with the surface roughness of tested samples. Seo and Yun [1] investigated the NO*_x_* removal rate of cement mortars containing TiO_2_ powers (85% anatase and 15% rutile) based on the TiO_2_ particle size and humidity. They [1] reported that smaller TiO_2_ particles are more effective for removing NO*_x_* than larger sized particles and the removal rate increases with increasing TiO_2_ mass replacement rate and decreasing humidity. Chen et al. [6] found that a TiO_2_-modified self-compacting mortar could effectively remove NO*_x_*, but the conversion of toluene through a photocatalytic reaction with the TiO_2_-modified mortar was not detected under typical outdoor conditions. Rhee et al. [7] considered two types of TiO_2_ powders, i.e., P-25 (a commercial rutile/anatase powder) and NP400 (a cheap anatase powder), for NO*_x_* removal by cement paste and mortars. The results showed that the two main factors important for enhancing the NO*_x_* removal efficiency are the density of the TiO_2_ powder and the surface roughness. The P-25 was more effective owing to its relatively lower apparent density of 3.41 g/mL as compared to the density of the NP400 of 3.70 g/mL, leading to a bulkier distribution in the cement composites; moreover, a higher surface roughness absorbs pollutants from air more efficiently. Interestingly, it was also observed that much higher NO*_x_* removal rates occurred at the bottom surfaces of the tested samples than at the casting (top) surfaces [7]. Guo and Poon [8] reported that white cement paste with incorporated nano-TiO_2_ exhibited a superior NO*_x_* removal capability compared to that of the ordinary Portland cement paste containing an identical amount of nano-TiO_2_ due to relatively abundant Fe_2_O_3_ in the latter. In addition, due to the much higher light absorption ability of ordinary Portland cement than that of white cement, fewer electron/hole pairs can be generated, which mitigates the capacity of TiO_2_ particles for NO*_x_* removal. A stronger charge transfer resistance in ordinary Portland cement also encourages more electron/hole pairs to recombine than separate, which also decreases the NO*_x_* removal capacity [7].

In addition to the photocatalytic action of TiO_2_ power in cement composites, several researchers [9,10,11] have investigated its impact on cement hydration, mechanical strength and shrinkage characteristics. Zhang et al. [9] reported that the use of nano-TiO_2_ increased the compressive strength of cement mortar as a result of the accelerated cement hydration and reduced pore sizes. It also decreased the drying shrinkage by reducing the water loss during drying owing to the reduced contact angle of the hardened cement paste caused by the TiO_2_ addition (greater hydrophilicity) [9]. Lee et al. [10] similarly reported that TiO_2_ nanoparticles accelerated the early age hydration and increased the compressive strength at higher dosages. The test results indicated that the size of the nanoparticles and the dispersibility were critical for the hydration rate; moreover, for a greater effect, smaller sized agglomerates were required [10]. The improvement in the durability of the concrete through the addition of TiO_2_ nanoparticles was reported by Jalal et al. [11] and Salemi et al. [12]. The chloride penetration depth in the cement composites, decreased by the addition of nanoparticles and the strength loss in composites containing 2% TiO_2_ by weight, was less than that in composites without added TiO_2_ under the action of freeze–thaw cycles. Likewise, the effectiveness of TiO_2_ powders or white cement for improving the NO*_x_* removal capability and various mechanical and durability properties of concrete has been extensively studied thus far.

Although concrete has been broadly used as a construction material for several decades, there remains a drawback awaiting a solution, i.e., the relatively low tensile strength and poor ductility. For this, conventional steel reinforcing bars or tendons have been contained in concrete structures for building construction, but due to a corrosion problem, there has been a frequently raised durability issue. To overcome its limitation, high-performance cement composites (HPCCs) containing discontinuous synthetic fibers have been developed [13,14,15,16,17,18]. Even though these are broadly used worldwide at present, no published studies are available concerning the effect of TiO_2_ powders on the properties of HPCCs with synthetic fibers. Accordingly, in this study, the influences of TiO_2_ powders and white cement on several properties of HPCCs containing varying amounts of glass and polyethylene (PE) fibers were analyzed, including their NO*_x_* removal ability, mechanical strength and volume stability. Two types of TiO_2_ powders (P-25 and NP400) were considered and a shrinkage-reducing admixture (SRA) was also incorporated to reduce the shrinkage strain and cracking potential from a practical point of view. For a rational evaluation of the observed test results, the microstructural changes in the cement mortar induced by the additives were evaluated using X-ray diffraction (XRD), and the variation in the surface tension of distilled water was analyzed according to the SRA.

## 2. Test Program

### 2.1. Properties of Special Ingredients

Two types of photocatalytic materials, i.e., the P-25 and NP400, were used with two different weight ratios of 5% and 10% and their transmission electron microscope (TEM) images are shown in Figure 1. The lower case “n” and “p” indicate the NP400 and P-25, respectively, and the subsequent numeral denotes their weight ratios of 5% and 10%. The P-25 was produced by spaying titanium chloride (TiCl_4_) into a kiln at temperatures ranging from 1000 °C to 1200 °C, while the NP400 was produced from titanium hydroxide (Ti(OH)_4_) in a vertical rotary kiln at a lower temperature of 600 °C for 4–5 h [7]. Even though the burning temperatures and production processes differed, the shapes and sizes of P-25 and NP400 were quite similar, as shown in Figure 1, coherent with the reports by Rhee et al. [7]. However, their crystal phases were different: the former was a rutile- and anatase-type powder, while the latter was an anatase-type powder because the crystallinity was crucial for the burning temperature. This means that the P-25 was more crystalline than its counterpart, which had a more amorphous structure.

To achieve volume stability, a glycol-based SRA (Dongnam Co., Ltd., Busan, Korea) with a density of 0.85 g/cm^3^ was added. This is a colorless liquid admixture without any solid content.

Two types of synthetic fibers, made of glass and PE, were also added to achieve the bridging capacity at crack planes, increasing the ductility of the concrete [17,18]. The geometric and physical properties are given in Table 1. A high-quality alkali-resistant glass fiber containing a high percentage of zirconia (ZrO_2_) ≥ 17% was supplied by Nippon Electric Glass Co., Ltd., Otsu, Japan and the lengths of the glass and PE fibers were 12.9 and 18.0 mm with densities of 2.80 and 0.96 g/cm^3^, respectively. As given in Table 2 and Figure 2, the diameter of the glass fiber was 2–3 times smaller than that of the PE fiber. Although their interfacial bond strengths were low owing to their hydrophobic nature, the PE fiber, with a much higher tensile strength > 2900 MPa, was primarily used to prevent a premature rupture failure mode that would negatively affect the post-cracking tensile performance of the composites, similar to previous studies [16,19].

### 2.2. Determination of Test Variables

To evaluate the influence of the synthetic fiber type and the amount of the shrinkage properties of the HPCCs, two types of fibers (glass and ultra-high molecular weight polyethylene (PE) [16]) and four volume fractions (0%, 0.5%, 1%, and 2%) were considered, as summarized in Table 2. From the shrinkage test results, an optimum fiber volume fraction of 1% was determined; the effect of the photocatalytic materials on the shrinkage properties was then analyzed. Two types of photocatalytic materials, P-25 and NP400, along with a white Portland cement, were incorporated by replacing a portion of the ordinary Type I Portland cement (OPC). It has been reported that the combined use of white cement and nano-TiO_2_ powder has a better NO*_x_* removal capability than OPC with the same amount of nano-TiO_2_ [8]. To reduce the shrinkage strain of the HPCCs, SRA was also utilized; to determine the optimum amount of SRA to be included, four different weight ratios to the binder materials were considered, ranging from 0% to 3%. A small amount, i.e., 1%, of the SRA was determined to be optimal for samples including photocatalytic materials based on the shrinkage strain measurements.

### 2.3. Mixture Proportion of the HPCC

The detailed mixture proportions of the HPCC are given in Table 3. OPC or white Portland cement, silica fume (SF), and class F fly ash (FA) were used as the cementitious materials. The chemical compositions and physical properties of the cementitious materials are summarized in Table 4. The OPC and white Portland cement have similar chemical compositions and Blaine fineness. The mean grain sizes of the OPC and SF are 22 and 0.31 μm, respectively. The class F FA has a Blaine fineness of 3117 cm^2^/g. For the fine aggregate, a combination of two different domestic sands #6 and #7 with mean grain sizes of approximately 294 and 174 μm, respectively, was used; a silica flour (S-SIL30) composed of more than 98% silicon dioxide (SiO_2_) with a mean grain size of 14.1 μm was used as a filler. A low water–binder (W/B) ratio of 0.28 was adopted. To overcome the poor workability of the mixture, a polycarboxylate superplasticizer (SP) with a density of 1.01 g/cm^3^ and a solid content of 30% was added. In addition, a viscosity agent was used for the fiber dispersibility and an antifoaming agent was used to provide excellent strength. The other ingredients, i.e., the TiO_2_ powders, SRA, and glass and PE fibers, were included in the standard mixture of the HPCCs in Table 3.

### 2.4. XRD Analysis

To evaluate the hydration products of HPCC based on the addition of SRA, photocatalytic materials and white Portland cement, XRD analyses were conducted using Cu Kα radiation at a wavelength of 1.5406 Å. To prevent an excessive SiO_2_ peak from the high amount of silica sand, cement paste was prepared for the XRD analyses without silica sand using the same ingredients and curing method. After curing the cubic samples of the cement paste, they were crushed to produce fine powders, passed through a standard 75 μm sieve, placed into a holder, and the surface was smoothed. The measurement range was 7°–70°.

### 2.5. Test Setup for Nitrogen Oxides (NO_x_) Removal

Figure 3 shows the detailed test setup for NO*_x_* removal. The NO*_x_* removal test was conducted according to standard ISO 22197-1 [20]. Thin HPCC panels with dimensions of 50 mm × 100 mm × 10 mm were used to evaluate the effect of adding the TiO_2_ powders and white Portland cement on the capacity for NO*_x_* removal, compared to that of the OPC. To analyze their net effect, both glass and PE fibers were excluded from the mixture used to make the panel samples. In these tests, an HPCC panel was inserted into the NO*_x_* removal machine (Figure 3) and exposed to 1 ± 0.015 ppmv nitric oxide (NO) gas at a flow rate of 3.0 L/min under UV light at 10 W/m^2^ for 2 h. During the test, the temperature and relative humidity in the chamber were maintained at 25 °C ± 2 °C and 50% ± 5%, respectively. Then, the NO gas flow was carried out in the dark for 30 min, after which the panel was UV-irradiated for 5 h. The NO removal rate was measured using a NO*_x_* analyzer (CM2041, Casella, UK) and a photometer (HD9021, Delta Ohm, Italy), and was calculated using the following equation:NO removal rate (%) = (*C_i_* − *C_eq_*)/*C_i_* × 100(1)
where *C_i_* is the initial NO concentration and *C_eq_* is the NO concentration after 5 h of UV irradiation.

### 2.6. Free Shrinkage Measurement

To measure the free shrinkage strain of the HPCCs, a number of prismatic molds with dimensions of 50 mm × 50 mm × 250 mm were prepared, as shown in Figure 4. Similarly to the recommendations of the Japan Concrete Institute (JCI) [21], a very thin Teflon sheet was placed inside the mold to prevent frictional restraint between the fresh HPCCs and inner surface of the mold. Then, a dumbbell-shaped and plastic-embedded strain gauge and a thermocouple were placed at the center of the mold using a nylon line because the self-consolidating nature of the HPCC material made it quite difficult to properly position the instruments after casting. All of the sensors were connected to a data logger system and the measurements were started before adding the concrete to obtain very early age strains. The fresh HPCCs were then placed into the molds and their exposed surface area was covered with a plastic sheet immediately to prevent evaporation. The shrinkage measurements were conducted in a room at constant temperature and humidity (*T* = 20 °C ± 0.5 °C and *RH* = 60% ± 5%, where *T* is the temperature and *RH* is the relative humidity). At 48 h after casting, the prismatic samples were demolded and exposed to the air in the room.

### 2.7. Mechanical Tests

At least three cylindrical specimens were fabricated and tested to measure the compressive strength of each variable. Similarly to the ultra-high-performance concrete (UHPC), a type of HPCC, the mechanical test samples were cured in a water tank with a high temperature of 90 °C for 48 h to promote strength development, as was recommended by the Federal Highway Administration (FHWA) in the US [22], and then tested. Specimens with a cross-sectional diameter of 100 mm and a height of 200 mm were used for the compressive strength measurements. To minimize any eccentric effect on the compressive strength, the casting surface was made flat using a diamond blade and a uniaxial load was monotonically applied with a universal testing machine (UTM). The compressive strength of the HPCC was then calculated using the following equation:*f_c_*′ = 4*P*_max_/*πd*^2^(2)
where *f_c_’* is the compressive strength, *P*_max_ is the maximum load applied and *d* is the diameter of the test cylinder.

To evaluate the tensile performance of the HPCC, a dog-bone-shaped sample was employed in accordance with the Japan Society of Civil Engineers (JSCE) recommendations [23]; these samples had cross-sectional dimensions of 13 mm × 30 mm and gauge length of 80 mm. Using a UTM with a maximum capacity of 250 kN, a uniaxial tensile force was applied monotonically and the tensile force was recorded by a load cell affixed to the test machine. To measure the elongation of the specimen under the tensile force, two linear variable differential transformers (LVDTs) were installed using an aluminum frame, as shown in Figure 5. The tensile stress was measured by dividing the recorded tensile force by the cross-sectional area; the tensile strain was obtained by dividing the displacement by the gauge length of 80 mm. To minimize the eccentric effect due to a secondary moment generated by asymmetrically formed initial matrix cracks, a fix-pin support condition was adopted, as recommended by Kanakubo [24]. Yoo et al. [25] reported that better fiber alignment to the direction of the tensile force was achieved with a smaller-sized specimen and it led to better flexural (or tensile) performance of the UHPC containing steel fibers. Due to the small sized cross-section of the dog-bone specimen, the tensile test results that were obtained in this study can be mainly used to compare the influences of test variables, i.e., fiber type and content, TiO_2_ powder, SRA and cement type, on the tensile behavior of HPCC.

## 3. Results and Discussion

### 3.1. Effects of TiO_2_ Powders and White Portland Cement on the NO Removal Capacity

As seen in Figure 6, the NO removal rate increased with the inclusion of 5% TiO_2_ powders, i.e., the P-25 and NP400, owing to the photocatalytic action under the UV irradiation. There was an obvious decrease of the NO concentration under the UV light and after turning off it, a re-increase was observed due to the continuous NO gas flows, similarly to a previous study by Seo and Yun [1]. The TiO_2_ absorbed a photon from the UV light and created an electron and hole pair. The NO was then transformed to a nitrate ion as a result of the existence of water and oxygen in the air, leading to a decrease in the NO concentration. For example, a removal rate of 1.12% was obtained by the OPC panel, while the removal increased to 3.11% and 3.88% with the addition of 5% P-25 and NP400, respectively. The NO removal capability was measured at the bottom surface of the test panels, and thus the use of the NP400 led to a better NO*_x_* removal capability than that of the P-25 owing to its higher density of 3.70 g/mL. There was a slight increase in the NO removal ability with the use of the white Portland cement instead of the OPC. However, the hybrid use of the white Portland cement and TiO_2_ powders provided better NO removal rates than that of the OPC with an identical amount (5%) of TiO_2_ powders. This is because the higher light absorption ability of the OPC leads to fewer electron and hole pairs, thus mitigating the NO removal capacity of the TiO_2_ particles [8]. Hence, the HPCC sample made of the white Portland cement with 5% NP400 provided the best NO removal capacity of approximately 4.46%. The NO removal capacity results demonstrated the necessity for developing novel methods to enhance the mechanical properties and the volume stability of HPCC containing TiO_2_ powders for practical applications.

### 3.2. Mechanical Properties

The compressive strengths of the tested samples are shown in Figure 7. The plain ordinary Portland cement (OP) samples had an average compressive strength of 83.4 MPa, which was generally higher than that of the OP-G series containing glass fibers. With the inclusion of glass fibers in the mixture, the compressive strength decreased by as much as 10% on average. This was consistent with the findings of Qureshi and Ahmed [26], who reported a decreased compressive strength of concrete with the inclusion of glass fibers at large volume fractions. The addition of PE fibers resulted in better compressive strengths than glass fibers at all volume fractions from 0.5% to 2% and produced similar compressive strengths on average to that of the plain OP sample. Similarly, Fanella and Naaman [27] noted that the addition of glass and polypropylene fibers did not provide any significant improvement in the compressive strength, in contrast to the addition of steel fibers. The advantages and disadvantages of adding discontinuous fibers on the compressive strength of concrete, e.g., the inhibition of crack formation and propagation and the non-uniform dispersion of fibers, have also been reported by Yoo et al. [28]. Banthia and Gupta [29] and Hsu and Hsu [30] also reported that the impact of the addition of fibers on the compressive strength was negligible. Thus, as compared to the effect of adding the glass and PE fibers on the post-cracking tensile behavior, it is obviously insignificant on the compressive strength of concrete due to their high aspect ratios and difficulty to perfect dispersion in the cement matrix, leading to deterioration in some samples.

The compressive strength of the HPCC including the 1% PE fibers by volume (OP-P1) was reduced by more than 2% with the addition of the SRA (Figure 7b). For example, the compressive strength increased slightly with the addition of the 1% SRA from 71.6 MPa to 75.6 MPa, whereas it decreased substantially to 53.4 MPa with the addition of the 2% and 3% SRA. This was consistent with the results of Folliard and Berke [31] for high-performance concrete (HPC) with and without silica fume; they reported that the 90-d compressive strength of HSC was reduced by roughly 9% with the addition of 1.5% SRA and the strength reduction was more obvious at the early age, which was mainly attributed to the impact of the SRA on delaying the early cement hydration. A continuous decrease in the compressive strength of the engineered cementitious composite (ECC) with an increase in the SRA content was also reported by Gao et al. [32].

In Figure 7c, the inclusion of TiO_2_ powders, i.e., the P-25 andNP400, generally reduced the compressive strength of the HPCC. For example, the compressive strength decreased by approximately 8% and 15% on average with the addition of the P-25 and NP400, respectively. Jimenez-Relinque et al. [5] also observed a decreased compressive strength of concrete with the addition of photocatalytic materials, even though they could have a positive filler effect [33], owing to the perceptual decrease in the amount of cement in the entire sample and the deteriorated homogeneity caused by the lower flowability. The decrease in the fluidity of the HPCC caused by adding photocatalytic materials was noticeable in this study, leading to a poorer homogeneity and a lower compressive strength. By replacing the OPC with white cement, a lower compressive strength was obtained; however, the effect of this replacement on the compressive strength was mitigated by adding the P-25 and NP400 simultaneously (Figure 7d).

The tensile stress and strain curves are shown in Figure 8. A total of six dog-bone specimens per each variable were tested and among them, one or two specimens exhibited crack localizations near the neck (beyond the gauge length). These samples were not used to obtain the average values, so that at least four specimens were adopted for obtaining the average results. It is clear that adding PE fibers was more effective for enhancing the post-cracking tensile performance of the HPCCs than adding glass fibers. Moreover, beyond a volume content of 1%, a strain-hardening response occurred, leading to a higher load-carrying capacity even after the formation of initial cracks in the matrix. By increasing the volume fraction of the PE fibers from 1% to 2%, the tensile performance was further improved, as shown in Figure 8a. On the other hand, a strain-softening response was still observed for the HPCCs with 2% glass fibers by volume. The poorer tensile performance observed in the HPCCs with glass fibers was mainly caused by their insufficient tensile strength, leading to premature ruptures and limiting the effective stress-carrying capacity at crack planes. As shown in Figure 9a, no glass fibers that had pulled out from the matrix were observed at the localized crack surface. However, owing to the superior tensile strength and hydrophobic surface of the PE fibers [34,35], they could be pulled out from the matrix without any breakage, thus continuously absorbing energy with a moderate crack-bridging capability. This is verified in Figure 9b showing that numerous PE fibers effectively bridged the localized crack plane. PE fiber was thus considered to be a more suitable reinforcement for HPCCs than glass fiber.

With the incorporation of the SRA, the tensile performance was deteriorated, as shown in Figure 8b. In addition, the strain-hardening response was changed to a strain-softening response, as in the glass-fiber-reinforced HPCC, owing to the reduced matrix shrinkage, which decreased the confining pressure of the fibers and the interfacial frictional resistance. The negative effect of the SRA addition on the tensile performance of various types of fiber-reinforced concrete (FRC) has been widely reported [32,36]. The addition of TiO_2_ powders, i.e., the P-25 and NP400, had no significant impact on the tensile performance of the HPCC (Figure 8c), whereas the use of the white Portland cement rather than the OPC produced a negative effect (Figure 8d). Thus, the HPCCs containing photocatalytic materials all exhibited strain-softening behavior, although the inclusion of 1% PE fibers effectively prevented brittle failure and improved the tensile ductility compared to the HPCCs without fibers. To achieve excellent tensile performance, hydrophilic or deformed synthetic fibers are recommended for HPCCs containing SRA or white Portland cement.

### 3.3. Free Shrinkage Behavior

#### 3.3.1. Effect of Synthetic Fiber Type and Amount

Several researchers [37,38] have reported that a proper determination of the time-zero point is important for a precise evaluation of concrete shrinkage, and this becomes more significant for HPC owing to its steep increase in early-age shrinkage strains [39]. Holt [37] noted that using the initial or final setting time as the time-zero for shrinkage measurement can be erroneous. Yoo et al. [38] suggested a novel method to determine the time-zero of UHPC as when inconsistent strain and internal temperature responses are initiated. By comparing the point where the measured strain behaved differently with the inner temperature to the developing time of tensile stress by restrained shrinkage tests, they [38] verified that this was close to the developing time of shrinkage stress. In this study, this point was thus used to eliminate initial deformation generated by temperature discrepancies, rather than desiccation, and to obtain the net shrinkage strain influencing the shrinkage cracking potential. Figure 10 summarizes the shrinkage behaviors of all of the tested HPCC samples. In Figure 10a, the plain OP series exhibited an initial sudden increase in the shrinkage strain to approximately 200 με, after which the rate of increase of the shrinkage strain was greatly reduced. At 48 h after casting, the shrinkage strain again began to increase substantially until approximately 8 d and converged to the ultimate strain value of approximately 1300 με. This shrinkage behavior indicated that the initial steep increase in the strain before 48 h was mainly due to autogenous shrinkage, and a significant portion of the second increase in the strain was due to drying shrinkage. The smallest amounts (0.5%) of glass and PE fibers were ineffective for reducing the shrinkage strains and similar 30-d shrinkage strains of approximately 1200 με were observed. On the other hand, the shrinkage strain of the plain HPCC was decreased with inclusion of glass and PE fibers at volume fractions greater than 1%, and the effectiveness of their addition for reducing the shrinkage increased with increasing volume content. This was consistent with the findings of previous studies [40,41,42], which reported a continuous decrease in the shrinkage strain of concrete with an increasing content of fibers up to approximately 3%. The reduced shrinkage was caused by the curbing of the shrinkage of the cement matrix through interfacial bonding between the fibers and the matrix. In addition, glass fiber was more effective for reducing the shrinkage strain than the PE fiber. For example, with the addition of 1% glass and PE fibers, the 28-d shrinkage strain of the HPCCs decreased by 7% and 4%, respectively. Zhang and Li [40] noted that steel and carbon fibers were more effective for reducing cement matrix shrinkage than polypropylene and polyvinyl alcohol fibers owing to their higher modulus. Thus, the higher Young’s modulus of the glass fiber yielded a higher efficiency than the PE fiber with a lower modulus. However, the difference between the degree of the shrinkage strain reduction with the glass and PE fibers was quite small because of the higher aspect ratio of the latter. Zhang and Li [40] also noted that fibers with a higher aspect ratio were more effective at reducing composite shrinkage than those with a lower aspect ratio.

#### 3.3.2. Effect of Shrinkage-Reducing Admixture (SRA)

The shrinkage behaviors of HPCCs with and without SRA are shown in Figure 10b. It is clear that adding SRA was effective for achieving the volume stability of the cement mortar. Generally, the effectiveness was increased with the increasing content of added SRA, owing to the decreased surface tension of the pore solution [42,43]. The main autogenous shrinkage mechanism at the early ages in this study is the self-desiccation of pore water, leading to the formation of a meniscus and pore pressure through surface tension. Thus, the substantial decrease in the surface tension of the pore solution incurred by adding the SRA suppressed the steep increase in the early age shrinkage strains before 48 h. In order to verify this explanation, the surface tension of distilled water with and without SRA was measured. The surface tension of the pure distilled water was found to be 0.0753 N/m and it decreased to 0.0315 and 0.0309 N/m with the adding of the SRA to the distilled water as much as 4% and 10%, respectively, corresponding to the 1% and 3% by weight of the binder (S1 and S3). This was consistent with the findings of Bentz et al. [44]. Figure 10b indicates that the initial shrinkage strain that developed up to approximately 200 με in the OP-P1 sample was eliminated by including the SRA, regardless of its amount. Furthermore, the drying shrinkage was also decreased by the addition of the SRA. This was the opposite of that observed with distilled water: the distilled water containing SRA led to faster mass loss at early ages than that without the SRA [44]. The decreased drying shrinkage of the cement mortar with the addition of the SRA can be explained by non-uniform drying throughout its thickness. Bentz et al. [44] reported that the initial drying at the exposed surface concentrated the SRA in the remaining pore solution, limiting the drawing of water from the pore solution with a higher surface tension beneath it. This reduced the mass loss rate of the cement mortar containing SRA. Owing to the decreased autogenous and drying shrinkage of the HPCCs, higher SRA contents resulted in greater reductions in the total shrinkage after 30 days (Figure 11). For example, the sample containing 3% SRA (OP-P1S3) produced a 30-d shrinkage strain of 698 με, which was only 57% of that of the OP-P1 sample without the SRA. Furthermore, by adding only a small amount of SRA, i.e., 1%, a significant decrease in the shrinkage strain was observed; for instance, the OP-P1S1 sample had a 34% reduction in shrinkage strain compared to that of the OP-P1.

#### 3.3.3. Effects of White Portland Cement and TiO_2_ Powders

To evaluate the influence of photocatalytic materials on the shrinkage behavior of HPCC, the shrinkage and time curves for the HPCCs with and without photocatalysis are shown in Figure 10c and d. In the case of commercial TiO_2_ power (Degussa, P-25), no noticeable impact on the shrinkage behavior was observed: the 30-d shrinkage strains of samples OP-P1S1-p5% and -p10% were 803 and 814 με, respectively, which were quite similar to that of sample OP-P1S1 (813 με). On the other hand, the addition of NP400, manufactured under a relatively low burning temperature, resulted in a slight decrease in the shrinkage strains: 779 and 727 με for samples OP-P1S1-n5% and -n10%, respectively, after 30 days. Therefore, it can be concluded that the shrinkage behavior of the HPCC was insignificantly or only slightly affected by the addition of the photocatalytic materials. By replacing the OPC with the white Portland cement, however, a significant decrease in the shrinkage strain from 813 με to 643 με was observed (Figure 11). This was inconsistent with the findings of Dellinghausen et al. [45], who reported that white Portland cement produced a higher total shrinkage compared to the grey OPC. In their study, higher compressive strengths were also observed with the use of the white Portland cement. Puertas et al. [46] measured the pore-size distribution of pastes made of white cement and OPC and discovered that although they exhibited quite similar pore-size distributions with age, the white cement paste produced more large-sized pores, ranging from 10 to 20 nm, than the OPC paste. Collins and Sanjayan [47] reported that the size of pores in which a meniscus was formed was a significant factor influencing the amount of drying shrinkage; under practical conditions, the shrinkage depends on the loss of water from mesopores with radii of 1.25–25 nm. It has been established [48] that pores with a smaller radius can result in larger capillary tensile forces at the meniscus, thus causing higher shrinkage strains. Furthermore, the white Portland cement somehow delayed the setting of the mortar compared to the OPC. Owing to the formation of larger-sized mesopores and the delayed setting, the HPCC samples made with white cement provided noticeably smaller total shrinkage than those made with the OPC. Owing to the insignificant influence of adding the P-25 and NP400, the hybrid use of the white Portland cement and the P-25 (or NP400) resulted in similar shrinkage behavior to that of the use of the white cement alone. The 30-d shrinkage strains for the hybrid samples with the P-25 and the NP400, were 643 and 678 με, respectively, which were close to that of white cement mortar only (643 με).

### 3.4. XRD Analysis

The XRD peak pattern was not affected by the existence and amount of SRA in Figure 12a, indicating that it did not influence the type of hydrates. As was reported in a previous study [38] that a delayed setting of UHPC was observed with the addition of the SRA, the XRD peaks of cement clinker, e.g., alite, belite and brownmillerite, were obviously obtained with the addition of SRA.

To verify the presence of the TiO_2_ powders, i.e., the P-25 and NP400, in the hardened cement paste, the corresponding XRD peak intensities were also analyzed. Rhee et al. [7] reported that the crystal phase of the commercial TiO_2_ power, P-25, was composed of both rutile and anatase, while the NP400, which is synthesized at lower temperatures, consisted purely of anatase. As illustrated in Figure 12b, for both of the samples containing the P-25 and the NP400, the XRD data revealed that the anatase peak with the highest intensity was observed at a 2θ of 25.37° [7,49]. Samples OP-P1S1-p5% and -p10% exhibited anatase peaks only, whereas samples OP-P1S1-n5% and -n10% included additional XRD peaks in addition to the highest peak intensity of anatase; the peak at a 2θ of 27.41° was the rutile crystal phase [7,49]. This verified the presence of the TiO_2_ powders in the composites. Furthermore, comparing the XRD peak patterns of the samples containing the TiO_2_ powders and the plain sample after the curing process revealed that the TiO_2_ powders did not influence the hydration process of the cementitious materials. Thus, their XRD peak patterns were quite similar to each other, excluding the anatase and rutile peaks.

The influences of the white Portland cement and the TiO_2_ powers on the XRD results are illustrated in Figure 12c. By replacing the OPC with white Portland cement, the XRD peaks of calcite and quartz were much higher than those for the OPC sample due to the different cement types. Similar to the OPC samples, rutile and anatase peaks were observed in the samples containing both the white Portland cement and the TiO_2_ powders (Figure 12c), indicating that the former did not influence the state of the TiO_2_ powders during the hydration process.

## 4. Conclusions

In this study, the mechanical and shrinkage properties of high-performance photocatalytic cementitious materials containing synthetic fibers and SRA were investigated. As the photocatalytic materials, the white Portland cement and two types of TiO_2_ powder (P-25 and NP400) were added, and glass and PE fibers were considered as the synthetic fibers. To analyze the results of the mechanical strength and volume change tests, XRD analyses were also conducted. Based on the test results and analyses, the following conclusions can be drawn:Using TiO_2_ powders effectively decreases the NO concentration, and the hybrid use of white Portland cement and TiO_2_ powders is most efficient.For identical volume fractions, the addition of PE fibers is more effective in terms of the mechanical properties, i.e., the compressive strength and tensile performance, than glass fibers in the HPCC mixture.Including the SRA and TiO_2_ powders deteriorates the compressive strength and tensile performance. The strain-hardening response of the plain HPCC containing 1% PE fibers is changed to a strain-softening response.Shrinkage of the HPCC could be reduced by adding glass and PE fibers at volume fractions of greater than 1%, and the effectiveness increases with the volume fraction. The glass fiber is more effective at reducing shrinkage strain than the PE fiber.Owing to the effectiveness of adding the SRA on decreasing both the autogenous and drying shrinkage, higher SRA contents result in greater reductions in the total shrinkage after 28 days. Approximately 43% of the total shrinkage of plain HPCC is reduced by adding 3% SRA.The TiO_2_ powders have no significant effect on the shrinkage reduction and hydration process of the HPCC. The use of white Portland cement instead of OPC is however effective in decreasing the shrinkage by as much as 20%.

## Figures and Tables

**Figure 1 materials-13-01828-f001:**
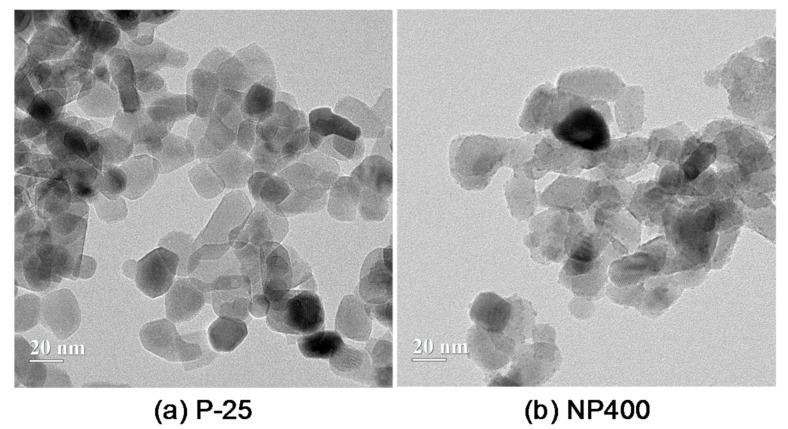
TEM images on the photocatalytic (titanium dioxide (TiO_2_)) materials: (**a**) the P-25 and (**b**) the NP400.

**Figure 2 materials-13-01828-f002:**
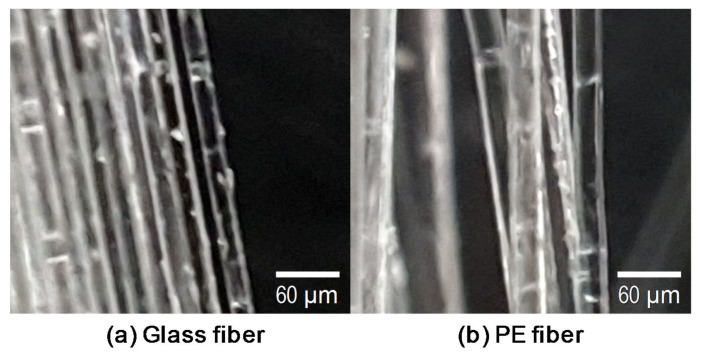
Pictures of (**a**) the glass fiber and (**b**) the polyethylene (PE) fiber.

**Figure 3 materials-13-01828-f003:**
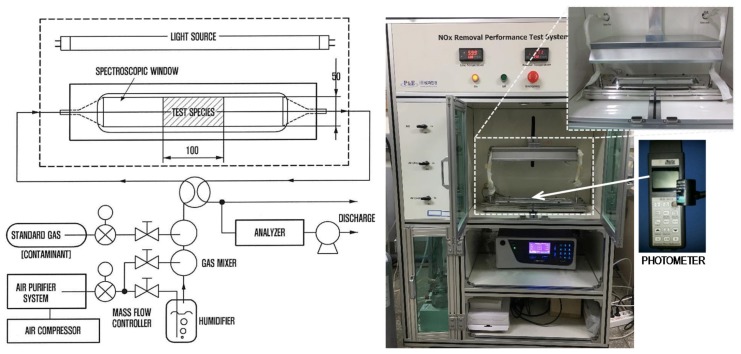
Test setup for nitrogen oxides (NO*_x_*) removal.

**Figure 4 materials-13-01828-f004:**
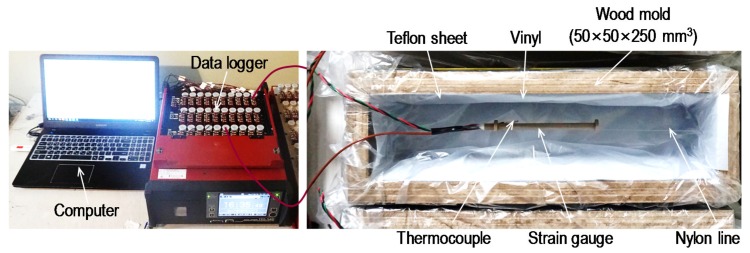
Test setup for the free shrinkage measurement.

**Figure 5 materials-13-01828-f005:**
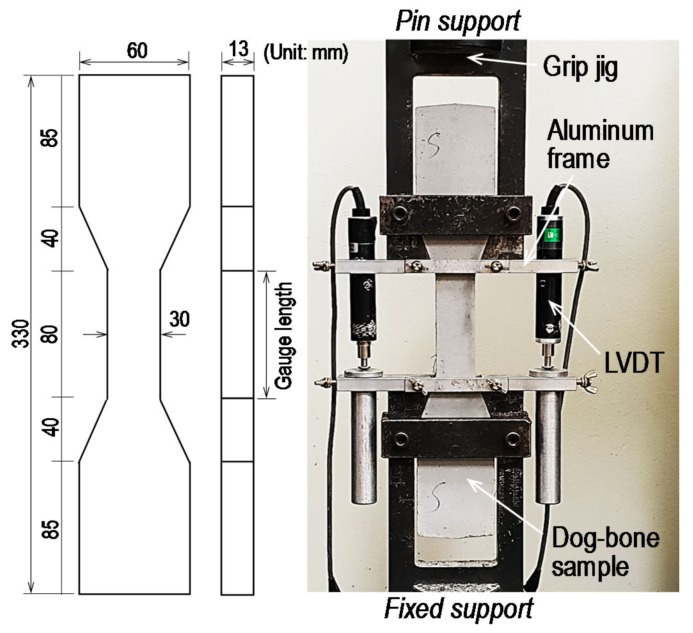
Direct tensile test setup.

**Figure 6 materials-13-01828-f006:**
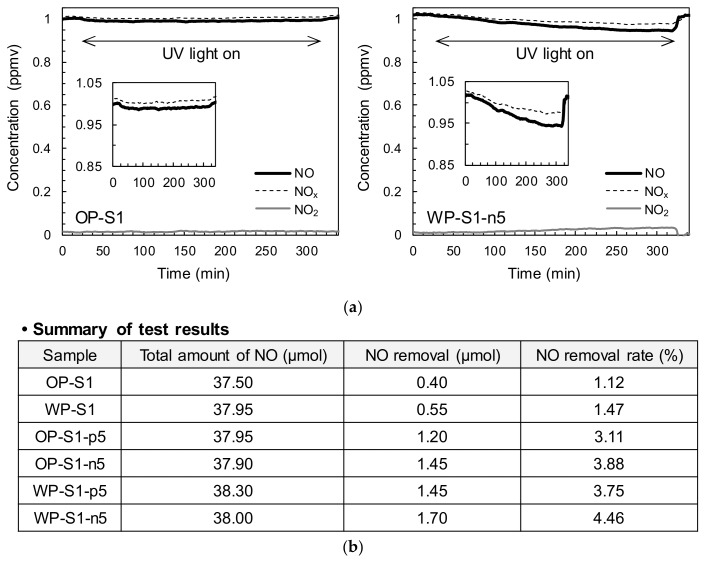
Summary of NO removal rates: (**a**) typical NO*_x_* concentrations and time behaviors (OP-S1 and WP-S1-n5), (**b**) summary of test results.

**Figure 7 materials-13-01828-f007:**
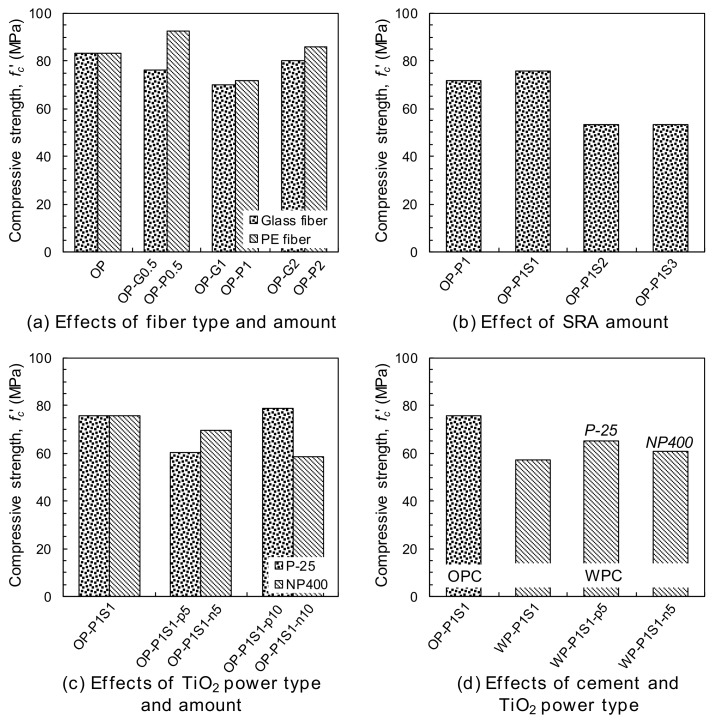
Summary of the compressive strength: (**a**) the effects of fiber type and amount, (**b**) the effect of the SRA amount, (**c**) the effects of the TiO_2_ powder type and amount, and (**d**) the effects of the cement and the TiO_2_ powder type.

**Figure 8 materials-13-01828-f008:**
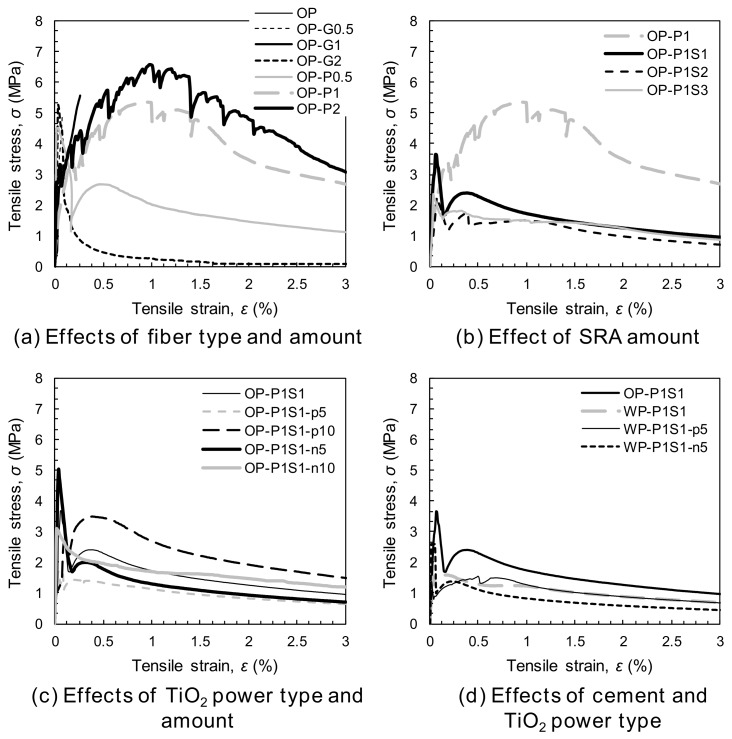
Summary of the tensile stress and strain curves: (**a**) the effects of the fiber type and amount, (**b**) the effect of the SRA amount, (**c**) the effects of the TiO_2_ powder type and amount, and (**d**) the effects of the cement and the TiO_2_ powder type.

**Figure 9 materials-13-01828-f009:**
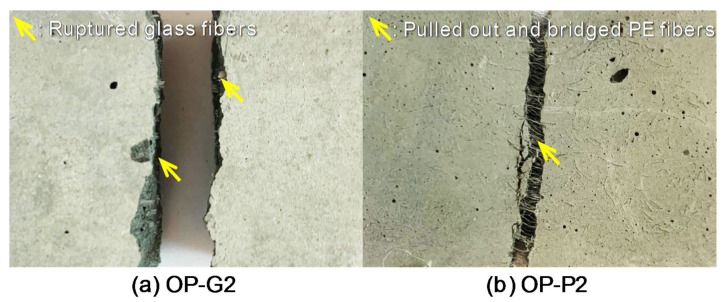
Pictures of the localized crack surface for the fiber failure mode: (**a**) OP-G2 and (**b**) OP-P2.

**Figure 10 materials-13-01828-f010:**
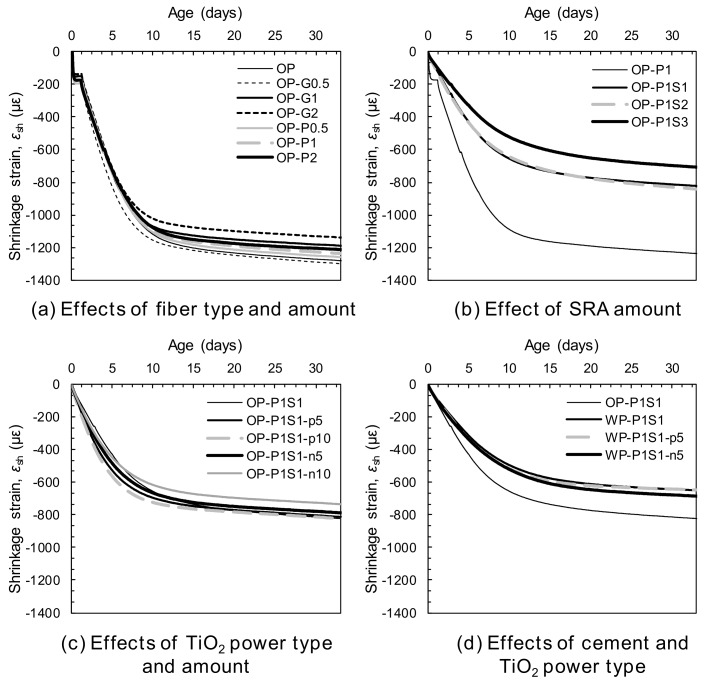
Shrinkage behaviors: (**a**) the effects of the fiber type and amount, (**b**) the effect of the SRA amount, (**c**) the effects of the TiO_2_ powder type and amount, and (**d**) the effects of the cement and the TiO_2_ powder type.

**Figure 11 materials-13-01828-f011:**
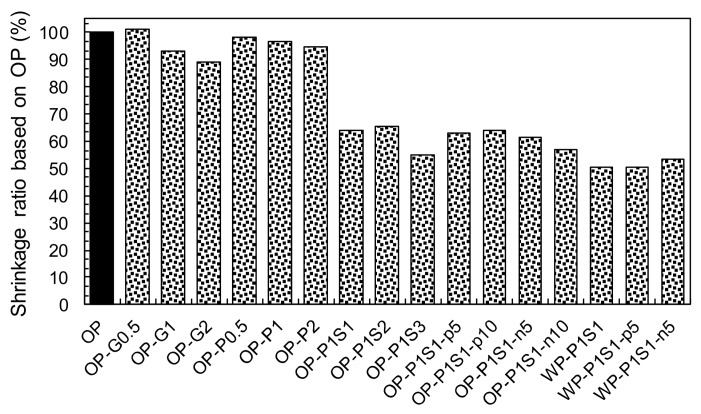
Summary of the 30-day shrinkage ratios based on the OP sample.

**Figure 12 materials-13-01828-f012:**
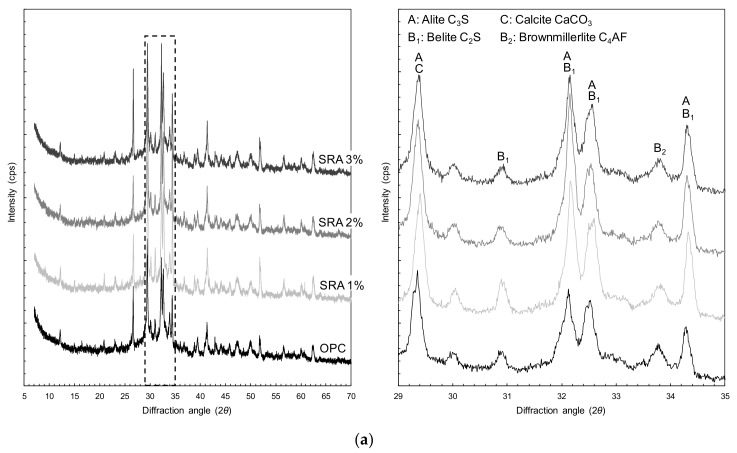
XRD patterns: (**a**) the effect of the SRA, (**b**) the effect of the TiO_2_ powders, and (**c**) the effect of the hybrid use of the white Portland cement and the TiO_2_ powders.

**Table 1 materials-13-01828-t001:** Properties of the glass and PE fibers.

Fiber Type	Diameter, *d_f_* (μm)	Length, *L_f_* (mm)	Aspect Ratio (*L_f_*/*d_f_*)	Density (g/cm^3^)	Tensile Strength (MPa)	Elastic Modulus (GPa)
Glass fiber	17.8 *	12.9	724.7	2.80	1500	74
PE fiber	30–32	18.0	562.5–600.0	0.96	2900–3160	88

PE fiber = polyethylene fiber; * Diameter of a filament.

**Table 2 materials-13-01828-t002:** Test variable.

Name	Volume Fraction of Fiber (%)	SRA (%)	TiO_2_ Powder * (%)	Note
OP	-	-	-	Control sample
OP-G0.5	0.5	-	-	Effects of fiber type and amount
OP-G1	1	-	-	
OP-G2	2	-	-	
OP-P0.5	0.5	-	-	
OP-P1	1	-	-	
OP-P2	2	-	-	
OP-P1S1	1	1	-	Effect of SRA amount
OP-P1S2	1	2	-	
OP-P1S3	1	3	-	
OP-P1S1-p5	1	1	5 (P-25)	Effects of TiO_2_ power type and amount
OP-P1S1-p10	1	1	10 (P-25)	
OP-P1S1-n5	1	1	5 (NP400)	
OP-P1S1-n10	1	1	10 (NP400)	
WP-P1S1	1	1	-	Effects of cement type and TiO_2_ power type
WP-P1S1-p5			5 (P-25)	
WP-P1S1-n5			5 (NP400)	

OP = ordinary Portland cement, G = glass fiber, P = polyethylene (PE) fiber, S = shrinkage-reducing admixture (SRA), p = Degussa TiO_2_ powder (P-25), n = TiO_2_ powder (NP400), WP = white Portland cement, and * weight ratio to binder.

**Table 3 materials-13-01828-t003:** Mix proportion of high-performance cement composites (HPCCs).

W/B	Mix Design (kg/m^3^)								
	Water	Cement *	Silica Fume	Fly Ash ^†^	Silica Flour **	Silica Sand	SP ^‡^	Viscosity Agent	Antifoaming Agent
0.28	249.46	707.33	70.73	141.47	141.47	848.80	11.12	3.33	2.00

W/B = water-to-binder ratio; SP = superplasticizer; * Type I ordinary Portland cement or white Portland cement are used; ^†^ Class F fly ash is used; ** Type of silica fume is S-SIL30.; ^‡^ Superplasticizer includes 30% solid and 70% water.

**Table 4 materials-13-01828-t004:** Chemical compositions and physical properties of the cementitious materials.

Composition % (Mass)	OPC	WPC	SF	FA
CaO	61.3	67.0	0.4	-
Al_2_O_3_	6.4	4.0	0.3	16.6
SiO_2_	21.0	22.0	96.0	3.8
Fe_2_O_3_	3.1	0.3	0.1	5.6
MgO	3.0	1.5	0.1	0.8
SO_3_	2.3	2.8	-	0.5
Specific surface (cm^2^/g)	3413	3400	200,000	3117
Loss ignition (%)	1.4	2.5	1.5	3.8
Density (g/cm^3^)	3.15	3.15	2.10	2.98

OPC = ordinary Portland cement, WPC = white Portland cement, SF = silica fume, and FA = fly ash.

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
