# Peer review of "High-Performance Photocatalytic Cementitious Materials Containing Synthetic Fibers and Shrinkage-Reducing Admixture"

_materials, 2020, doi:10.3390/ma13081828_

Round 1

Reviewer 1 Report

The paper investigated the NOx removal ability, mechanical and shrinkage properties of high-performance cement composites adding by fibers and SRA. Before this manuscript can be accepted, some points need to be further explained and modified, which are shown as follows:

  1. Introduction section, the form of superoxide and nitrate ion is irregular.
  2. In the section of Test Program, Table 1 is not clear. What is the TiO2 powder %? Does it add by mass ratio or volume ratio? The abbreviation of “n5” “p5” should be explained. In addition, the column of “cement type” “fiber type” seems to be unnecessary.
  3. Section 2.3 is about the raw materials, which is suggested to be placed at the beginning of the Test Program.
  4. Is equation (1) correct? Should the denominator be the initial NO concentration?
  5. The starting point of shrinkage should be further explained, there actually is a standard test method for autogenous and drying strain measurement of cementitious materials. The method used in the paper should be explained.
  6. The curing age of the mechanical property test should be mentioned.
  7. In section 3.1, the effect of Fe2O3 on NO removal capacity should have a detailed explanation.
  8. In figure 6, test results were presented in the form of both table and figure. Please avoid it.
  9. In figure 6, how to explain the concentration of NO rises at the end of 300min of WP-S1-n5?
  10. In figure 7, labels of abscissa are suggested to use the abbreviation in line with table 1.
  11. In section 3.2, mechanical test results should be explained more comprehensively. Why does the compressive strength of cement composites with PE fiber increase at 0.5% and 2% dosage but decreased at 1% dosage?
  12. In section 3.3.2, the mechanism of SRA is mainly due to the capillary tension. In some research papers, the delay of setting time is the harmful effect of SRA. Why the author thinks the effectiveness of SRA is related to the delay of setting time? If it is an important parameter, then the test of setting time should be added.
  13. In section 3.4, XRD analysis should be modified. Actually, the peak intention of XRD cannot explain the content of phase. It is also associated with crystallinity and some other factors.

Author Response

Reviewer #1

The paper investigated the NOx removal ability, mechanical and shrinkage properties of high-performance cement composites adding by fibers and SRA. Before this manuscript can be accepted, some points need to be further explained and modified, which are shown as follows:

Answer: First, thank you so much for your valuable and useful comments on our paper. We have carefully considered all your comments. The revised manuscript is attached for your reconsideration. We would like to thank you for your comment which significantly improved the quality of our paper.

  1. Introduction section, the form of superoxide and nitrate ion is irregular.

Answer: As you recommended, the form of superoxide and nitrate ion is modified.

  1. In the section of Test Program, Table 1 is not clear. What is the TiO2 powder %? Does it add by mass ratio or volume ratio? The abbreviation of “n5” “p5” should be explained. In addition, the column of “cement type” “fiber type” seems to be unnecessary.

Answer: As you mentioned, the TiO2 was added by the weight ratio to total binder. Thus, this is newly added below Table 1.

  In addition, to give the meaning of “n5” and “p5”, the following explanations are added in the text.

“… were used with two different weight ratios of 5 and 10%, and their transmission electron microscope (TEM) images are shown in Fig. 1. The lower case ‘n’ and ‘p’ indicate the NP400 and P-25, respectively, and the subsequent numeral denotes their weight ratios of 5 and 10%.”

As you recommended, the columns of “cement type” and “fiber type” are now deleted.

  1. Section 2.3 is about the raw materials, which is suggested to be placed at the beginning of the Test Program.

Answer: As you recommended, section 2.3 is now moved to the beginning of the Table Program.

  1. Is equation (1) correct? Should the denominator be the initial NO concentration?

Answer: Thank you! That was a misprint, and now it is corrected.

  1. The starting point of shrinkage should be further explained, there actually is a standard test method for autogenous and drying strain measurement of cementitious materials. The method used in the paper should be explained.

Answer: As you recommended, the starting point of shrinkage measurement is additionally explained in the text, as follows.

“By comparing the point where the measured strain behaved differently with the inner temperature to the developing time of tensile stress by restrained shrinkage tests, they [38] verified that this is close to the developing time of shrinkage stress.”

As you mentioned, we adopted a test method of autogenous shrinkage given by JCI. Thus, the following explanations are added in the text to give this information to readers with an additional reference1.

“In similar to the recommendations of Japan Concrete Institute (JCI) [21], a very thin Teflon sheet was placed inside the mold to prevent frictional restraint between the fresh HPCC and inner surface of the mold.”

-----------------------------------------------------------------------------------------------------------------------------

  1. Japan Concrete Institute. Committee report. Autogenous shrinkage of concrete. In: Tazawa E, editor, E&FN Spon; 1999. p. 3-62.

  1. The curing age of the mechanical property test should be mentioned.

Answer: As was recommended by FHWA in US2 for UHPC, the mechanical test samples were cured in a water tank with a high temperature of 90 °C for 48 h to promote strength development.

As you recommended, the following sentence is now added in the text.

“In similar to ultra-high-performance concrete (UHPC), a type of HPCC, the mechanical test samples were cured in a water tank with a high temperature of 90 °C for 48 h to promote strength development as was recommended by FHWA in US [22] and then tested.”

-----------------------------------------------------------------------------------------------------------------------------

  1. Graybeal BA. Material property characterization of ultra-high performance concrete (No. FHWA-HRT-06-103), 2006.

  1. In section 3.1, the effect of Fe2O3 on NO removal capacity should have a detailed explanation.

Answer: This is not clear yet and controversial. Thus, we now delete the part that you pointed out, as follows.

“… (1) the OPC contains a higher amount of Fe2O3 than white Portland cement, and (2) …”

  1. In figure 6, test results were presented in the form of both table and figure. Please avoid it.

Answer: Graphs in Fig. 6 are for only two samples (OP-S1 and WP-S1-n5) to explain the comparative behaviors of NOx removal according to the existence of TiO2 powder. In addition, all of the test results are summarized in the bottom table. So, we believe that the current form is better than removing the figure because if it is removed, it is quite difficult to explain the NOx removal behavior according to UV light and time. Please consider this comment again carefully.

  1. In figure 6, how to explain the concentration of NO rises at the end of 300min of WP-S1-n5?

Answer: This is because the NO gas flows continuously into the chamber even after turning off the UV light. Thus, rises of NO concentration is generally observed in previous studies3.

As you recommended, in order to help readers’ understanding, the following sentence is newly added in the text, along with the additional reference3.

  “There was an obvious decrease of the NO concentration under the UV light, and after turning off it, a re-increase of it was observed due to the continuous NO gas flows, similar to a previous study by Seo and Yun [1].”

-----------------------------------------------------------------------------------------------------------------------------

  1. Seo, D.; Yun, T.S. NOx removal rate of photocatalytic cementitious materials with TiO2 in wet condition. Build. Environ. 2017, 112, 233-240.
  2. In figure 7, labels of abscissa are suggested to use the abbreviation in line with table 1.

Answer: As you recommended, the labels in Figure 7 is now modified.

  1. In section 3.2, mechanical test results should be explained more comprehensively. Why does the compressive strength of cement composites with PE fiber increase at 0.5% and 2% dosage but decreased at 1% dosage?

Answer: It is well known that the contribution of adding fibers on the compressive strength of concrete is minor4,5. Especially, due to very high aspect ratios of glass and PE fibers used, it was quite difficult to perfectly disperse them in the cement matrix, leading to deterioration in some samples. Thus, along with a benefit of adding fibers limiting crack formation, there was a similar or slightly smaller compressive strengths obtained in the samples including the fibers.

As you mentioned, to help readers’ understanding on the compressive strength more comprehensively, the following sentences are added in the text.

“Thus, as compared to the effect of adding the glass and PE fibers on the post-cracking tensile behavior, it is obviously insignificant on the compressive strength of concrete due to their high aspect ratios and difficulty to perfect dispersion in the cement matrix, leading to deterioration in some samples.”

-----------------------------------------------------------------------------------------------------------------------------

  1. Banthia, N.; Gupta, R. Hybrid fiber reinforced concrete (HyFRC): fiber synergy in high strength matrices. Mater. Struct. 2004, 37 (10), 707-716.
  2. Hsu, L.S.; Hsu, C.T.T. Stress–strain behavior of steel-fiber high-strength concrete under compression. ACI Struct. J. 1994, 91 (4), 448-457.

  1. In section 3.3.2, the mechanism of SRA is mainly due to the capillary tension. In some research papers, the delay of setting time is the harmful effect of SRA. Why the author thinks the effectiveness of SRA is related to the delay of setting time? If it is an important parameter, then the test of setting time should be added.

Answer: As you pointed out, we now delete the effectiveness of SRA in terms of the delayed setting, as follows.

“… resulting delayed setting of the mortar [33] and …”

Unfortunately, we didn’t measure the setting time. However, since we now delete the sentence that you mentioned, the setting property is now mentioned in the text entirely.

  1. In section 3.4, XRD analysis should be modified. Actually, the peak intention of XRD cannot explain the content of phase. It is also associated with crystallinity and some other factors.

Answer: As you mentioned, the Bragg’s law based XRD analysis is a qualitative analysis to determine existence of some unknown phases based on the XRD peak patterns. Therefore, some sentences that were related to the peak intensity of phase content are now deleted, as follows.

“By including the SRA, slightly higher peaks corresponding to alite, belite, and brownmillerite were observed compared to the plain sample, as shown in Fig. 12a.”

“… and its intensity seems to increase with increasing amount of TiO2 powders …”

“… higher peaks corresponding to cement clinker were obtained. In particular, …”

In addition, as you recommended the section ‘3.4 XRD Analysis’ is now significantly modified and improved.

We sincerely appreciate your useful comments on our paper again. We did our best to address all your comments. Please kindly and carefully take into account our answers above and reconsider your decision.

Reviewer 2 Report

The beneficial effects of photocatalytic materials like the titanium dioxide on concretes have long been known. In this context, the paper concers the effects of TiO2 powders on the properties of high-performance cement composites (HPCC), in which shrinkage-reducing admixtures (SRA) and synthetic fibers was also incorporated. The combined effects of these three components in HPCCs have been investigated in detail from an experimental point of view.

The paper is certainly of interest to the readers of the Journal, it is well written and the results obtained are of good quality. Therefore, this reviewer proposes to accept the paper for publication.

The authors may consider whether to cite the following two papers:

  1. L. Lanzoni , A. Nobili, A.M. Tarantino. Performance evaluation of a polypropylene-based draw-wired fibre for concrete structures. Construction and Building Materials, vol. 28, 2012, pp. 798-806
  2. A. Nobili, L. Lanzoni , A.M. Tarantino. Experimental investigation and monitoring of a polypropylene-based fiber reinforced concrete road pavement. Construction and Building Materials, vol. 47, 2013, pp. 888-895

pioneers in the use of synthetic fibers in concretes.

Author Response

Reviewer #2

The beneficial effects of photocatalytic materials like the titanium dioxide on concretes have long been known. In this context, the paper concerns the effects of TiO2 powders on the properties of high-performance cement composites (HPCC), in which shrinkage-reducing admixtures (SRA) and synthetic fibers was also incorporated. The combined effects of these three components in HPCCs have been investigated in detail from an experimental point of view.

The paper is certainly of interest to the readers of the Journal, it is well written and the results obtained are of good quality. Therefore, this reviewer proposes to accept the paper for publication.

Answer: First, thank you so much for your valuable and useful comments on our paper. We have carefully considered all your comments. The revised manuscript is attached for your reconsideration. We would like to thank you for your comment which significantly improved the quality of our paper.

The authors may consider whether to cite the following two papers:

  1. Lanzoni, A. Nobili, A.M. Tarantino. Performance evaluation of a polypropylene-based draw-wired fibre for concrete structures. Construction and Building Materials, vol. 28, 2012, pp. 798-806
  2. Nobili, L. Lanzoni , A.M. Tarantino. Experimental investigation and monitoring of a polypropylene-based fiber reinforced concrete road pavement. Construction and Building Materials, vol. 47, 2013, pp. 888-895

pioneers in the use of synthetic fibers in concretes.

Answer: As you recommended, the above two papers are additionally quoted in the text.

Reviewer 3 Report

The submitted article “materials-774640” entitled “High-Performance Photocatalytic Cementitious Materials Containing Synthetic Fibers and Shrinkage-Reducing Admixture” is an original and interesting experimental investigation of the mechanical, shrinkage and chemical properties of photocatalytic cementitious materials containing synthetic fibers and a shrinkage-reducing admixture. It is an object which is still open to question since the existing published work in this field of study is rather limited. The topic of the paper fits within the scope of the Journal. Figures are precise and helpful. The manuscript is well-structured and well-written. The following minor comments are raised for authors’ reference:

- In introduction, the literature review is rather informative but could be improved providing more convincing motivations of this research. Although the tasks and the research significance of the study are clearly defined, it is recommended to be highlighted. The main objectives of this work are also briefly stated and seem rather modest since important conclusion have been derived from the performed tests.

- Concerning the direct tensile tests, the number of the “dog-bone” shaped specimens that were fabricated and tested to measure the tensile performance of each high-performance cement composite sample is not clarified. The location of the final crack and failure during these tests should be reported, too.

- The width of the tensile specimens is rather low (13 mm). This seems a parameter that could influence the results since the cross-section under tension is 13/30 mm. Some comments concerning this issue would be useful.

Author Response

Reviewer #3

The submitted article “materials-774640” entitled “High-Performance Photocatalytic Cementitious Materials Containing Synthetic Fibers and Shrinkage-Reducing Admixture” is an original and interesting experimental investigation of the mechanical, shrinkage and chemical properties of photocatalytic cementitious materials containing synthetic fibers and a shrinkage-reducing admixture. It is an object which is still open to question since the existing published work in this field of study is rather limited. The topic of the paper fits within the scope of the Journal. Figures are precise and helpful. The manuscript is well-structured and well-written. The following minor comments are raised for authors’ reference:

Answer: First, thank you so much for your valuable and useful comments on our paper. We have carefully considered all your comments. The revised manuscript is attached for your reconsideration. We would like to thank you for your comment which significantly improved the quality of our paper.

- In introduction, the literature review is rather informative but could be improved providing more convincing motivations of this research. Although the tasks and the research significance of the study are clearly defined, it is recommended to be highlighted. The main objectives of this work are also briefly stated and seem rather modest since important conclusion have been derived from the performed tests.

Answer: Thank you for this valuable comment on our manuscript. We agree to you that there is no clear reason and main object of this study in the INTRODUCTION. Therefore, the following sentences are newly added in the text along with a citation of more references1,2.

“Likewise, the effectiveness of TiO2 powders or white cement for improving the NOx removal capability and various mechanical and durability properties of concrete has been extensively studied thus far.

Although concrete has been broadly used as a construction material for several decades, there remains a drawback awaiting solution, i.e., relatively low tensile strength and poor ductility. For this, conventional steel reinforcing bars or tendons have been contained in concrete structures for building construction, but due to a corrosion problem, there has been a durability issue raised frequently. To overcome its limitation, high-performance cement composites (HPCC) containing discontinuous synthetic fibers has been developed [13-18]. Even though these are broadly used worldwide at present, no published studies are available concerning the effect of TiO2 powders on the properties of HPCC with synthetic fibers.

-----------------------------------------------------------------------------------------------------------------------------

  1. Lanzoni, L.; Nobili, A.; Tarantino, A.M. Performance evaluation of a polypropylene-based draw-wired fibre for concrete structures. Constr. Build. Mater. 2012, 28, 798-806.
  2. Nobili, A.; Lanzoni, L.; Tarantino A.M. Experimental investigation and monitoring of a polypropylene-based fiber reinforced concrete road pavement. Constr. Build. Mater. 2013, 47, 888-895.

- Concerning the direct tensile tests, the number of the “dog-bone” shaped specimens that were fabricated and tested to measure the tensile performance of each high-performance cement composite sample is not clarified. The location of the final crack and failure during these tests should be reported, too.

Answer: We fabricated six dog-bone specimens per each variable, and among them one or two specimens exhibited crack localization at near the neck (beyond the gauge length). These samples were not used to draw figures an obtain average values.

As you recommended, the following sentences are now added in the text to give this information.

A total of six dog-bone specimens per each variable were tested, and among them, one or two specimens exhibited crack localization at near the neck (beyond the gauge length). These samples were not used to obtain the average values, so that at least four specimens were adopted for obtaining the average results.”

- The width of the tensile specimens is rather low (13 mm). This seems a parameter that could influence the results since the cross-section under tension is 13/30 mm. Some comments concerning this issue would be useful.

Answer: We conducted the direct tensile test according to the JSCE recommendations. However, we totally agree with you that there might be some concerns in terms of such a small-sized cross-section. The most significant effect will be a degree of fiber alignment. Yoo et al.3 verified the better fiber alignment observed in small-sized specimens, leading to better flexural (or tensile behavior).

  As you recommended, in order to let readers’ know about the issue of such a small dimension that leads to better behavior, the following sentences are newly added in the text along with citing an additional reference1.

Yoo et al. [25] reported that better fiber alignment to the direction of tensile force is achieved at a smaller-sized specimen and it leads to better flexural (or tensile) performance of UHPC containing steel fibers. Due to the small-sized cross-section of dog-bone specimen, the tensile test results that are obtained in this study can be mainly used to compare the influences of test variables, i.e., fiber type and content, TiO2 powder, SRA, and cement type, on the tensile behavior of HPCC.

-----------------------------------------------------------------------------------------------------------------------------

  1. Yoo, D.Y.; Banthia, N.; Kang, S.T.; Yoon, Y.S. Size effect in ultra-high-performance concrete beams. Eng. Fract. Mech. 2016, 157, 86-106.

We sincerely appreciate your useful comments on our paper again. We did our best to address all your comments. Please kindly and carefully take into account our answers above and reconsider your decision.

Round 2

Reviewer 1 Report

The authors have revised the original manuscript, and it can be accepted.